# Quantitative Assessment on Optical Properties as a Basis for Bioluminescence Imaging: An Experimental and Numerical Approach to the Transport of Optical Photons in Phantom Materials

**DOI:** 10.3390/s23146458

**Published:** 2023-07-17

**Authors:** Tim Karstens, Theresa Staufer, Rasmus Buchin, Florian Grüner

**Affiliations:** Institut für Experimentalphysik and Center for Free-Electron Laser Science (CFEL), Universität Hamburg, Luruper Chaussee 149, 22761 Hamburg, Germany; tim.karstens@cfel.de (T.K.);

**Keywords:** bioluminescence imaging, Geant4 simulations, optical physics, optical properties, soft tissue

## Abstract

Bioluminescence imaging (BLI) is a widely used technique in preclinical scientific research, particularly in the development of mRNA-based medications and non-invasive tumor diagnostics. It has become an essential tool in current science. However, the current state of bioluminescence imaging is primarily qualitative, making it challenging to obtain quantitative measurements and to draw accurate conclusions. This fact is caused by the unique properties of optical photons and tissue interactions. In this paper, we propose an experimental setup and Geant4-simulations to gain a better understanding of the optical properties and processes involved in bioluminescence imaging. Our goal is to advance the field towards more quantitative measurements. We will discuss the details of our experimental setup, the data we collected, the outcomes of the Geant4-simulations, and additional information on the underlying physical processes.

## 1. Introduction

Bioluminescence imaging (BLI) has emerged as a vital imaging technique in modern scientific research, playing a crucial role in the development of mRNA-based medications and the preclinical non-invasive tumor diagnostics [1,2]. BLI allows for the visualization of biological processes in real-time, utilizing the light emitted by bioluminescent molecules [3]. However, despite its widespread use, BLI is primarily qualitative in nature; hence, obtaining quantitative measurements can be challenging due to the inherent properties of optical photons and tissue [4]. Progress in the field of BLI has been made on the one hand at the technical level, notably through the development of tomographic procedures and the correlation of these with CT and MRI images [5,6]. On the other hand, reconstruction methods that aim to solve the inverse problem, either mathematically or, more recently, using artificial intelligence [7,8], represent another area of new developments. These methods, however, are based on assumptions regarding optical parameters and anticipated tissue behavior. Our research reveals that some optical processes are indistinguishable from each other, and the arrangement of different tissue types significantly impacts the recorded signals.

To address these challenges and limitations, we have developed an abstract model of bioluminescence to better understand the fundamental physics of optical photon behavior in soft tissue. In our experiments, we utilized an LED that emits light at the same wavelength as the commonly used luciferase/luciferin reaction [9], enabling us to replicate the photon source of BLI effectively. Using this model, we simulated photon propagation in phantom materials using Monte Carlo methods. Our goal is to improve the quantification of bioluminescence imaging and to predict its behavior under different experimental conditions.

This paper takes a deeper look into the physics and simulations to advance BLI towards quantifiable measurements. We utilized Geant4-simulations to dissect the optical properties and processes involved in BLI [10]. Geant4 was selected for several reasons. Firstly, it is an open-source project, which enables numerous research groups utilizing the software. Additionally, a large community has already developed and implemented various physical processes and detectors. An abundance of sample codes is available for comprehensive data analysis, perfectly aligning with our purpose [11]. Moreover, for the fundamental application our research focuses on, the use of Geant4 alone suffices. The Geant4 Application for Emission Tomography (GATE) extension, commonly utilized in numerous bioluminescence publications, provides additional capabilities for tomographic applications that are not required in our study [12,13,14]. By comparing experimental data with simulation results, we aim to provide valuable insights into the underlying physical processes and to improve the accuracy and reliability of BLI measurements [15].

In the following sections, we will discuss in detail the experimental setup, the data acquired from our experiments, and the Geant4-simulation. Additionally, we will present supplementary data on the physical processes involved in BLI. This comprehensive analysis will contribute to a better understanding of the challenges and opportunities regarding quantification but will also show the intrinsic limitations of this imaging technique.

## 2. Materials and Methods

### 2.1. Experimental Setup

Our study revolves around a relatively simple setup at its core, comprising a CMOS camera, a system of tubes with an aperture, and an LED serving as the light source. The material under investigation consisted of polyurethane, titanium dioxide (TiO_2_), and carbon black as an absorber molecule. The phantom material, designed specifically for bioluminescence imaging to mimic the optical properties of soft tissue [16], was originally developed by INO and later obtained in its final form from MediLumine [17].

To ensure simplicity and minimize unknown variables, we intentionally designed the setup to be as straightforward as possible, allowing us to create a Geant4-simulation setup with limited or no ambiguity. We discuss the various components used in detail below, and Figure 1 illustrates the configuration and Section 2.9 describes the simulated setup in comparison. A schematic representation of the structure and path of the light is shown as a flow chart in Figure 2.

### 2.2. Camera

The Basler aca1920-40gm (Basler AG, Ahrensburg, Germany) camera features a monochrome Global Shutter CMOS sensor (Sony IMX249LLJ) with a resolution of 1920 × 1080 pixels and a pixel size of 5.86 μm. It has a high quantum efficiency (QE) of 70%, making it highly sensitive to light for low-light imaging applications. The low mean dark current of 6.7 electrons per pixel per second results in clean and noise-free images even in long-exposure scenarios [18].

### 2.3. Phantom Materials

As mentioned earlier, the phantom material used in our study is composed of polyurethane mixed with a specific amount of a scattering agent consisting of TiO_2_, along with an absorbing agent, namely, carbon black [19]. We chose this material to ensure that our simulations have a well-defined and properly characterized phantom. MediLumine kindly provided us with characterization data specifically for the desired wavelength of 590 nm. The results we obtained were as follows: 9.81 ${cm}^{-1}\;$ for ${µ}_{s}^{\prime }$, 10.82 ${cm}^{-1}$ for ${µ}_{a}$ and 0.62 for the anisotropy factor g [19]. We used small platelets of this phantom material fitting exactly into the inner spacer ring in front of the camera. Furthermore, we also examined another material consisting of 3D-printed polymethyl methacrylate (PMMA), which is commonly used in radiology research for creating mouse phantoms [20]. We used small platelets of this PMMA material with two different thicknesses: 2 mm and 5 mm. These platelets were carefully positioned between the camera sensor and the pinhole aperture to capture clear images of the transmitted beam passing through the phantom material.

### 2.4. Tube/Aperture System

To obtain a precisely defined light beam, we developed a workpiece (Figure 3) that contained an aperture of 1 mm and also served as a connector between the camera mount and a tube system (Thorlabs SM1L40). This allowed us to create a self-contained system that prevented both the entry and exit of light.

### 2.5. Light Source

We used an LED (Kingbright L934SYD) as a light source because of its simplicity of use and its well-known optical parameters [21]. The LED was chosen based on the wavelength of the emitted light, its radiation characteristics, and its brightness. The dominant wavelength is 588 nm, the peak wavelength is 590 nm and the spectral line half-width is 35 nm. The desired wavelength was within the range of common bioluminescence systems to enable good transferability [9,22]. The LED was operated with a self-designed current source, which allows a constant current and a monitoring of the actually flowing current during the measurements.

### 2.6. Image Acquisition

Images of the transmitted beam through the phantom material were acquired using the CMOS camera. The camera was triggered to capture images of the beam for each thickness of the phantom material. The images were taken with 8-bit monochrome. Camera exposure time was adjusted individually, depending on the thickness of the phantom material, and set to a maximum gray value of 150 to avoid false images and overexposure. Multiple images were taken to ensure reproducibility and to account for any potential variability in the beam intensity.

### 2.7. Image Analysis

Python-based image analysis scripts were developed to process the acquired images. The scripts were designed to extract the radial profile, line profile, and estimated the total number of photons from the images. The radial profile was obtained by averaging the intensity values along concentric circles centered at the pinhole aperture. The line profile was extracted by averaging the intensity values along a straight line passing through the center of the pinhole aperture. The estimated total number of photons was calculated by integrating the intensity values over the entire image. The line and radial profile were further analyzed and the full width half maximum (FWHM) value was calculated in mm to obtain a quantitative measurement of the signal broadening. Additionally, we calculated the mean error as an indicator of agreement. This error was obtained by dividing the sum of values on the y-axis by the total number of values on the x-axis.

### 2.8. Geant4-Simulation

Geant4 is an open-source toolkit, primarily written in C++, designed for simulating the passage of particles through matter, used extensively in fields like high energy physics and medical physics. Its open-source characteristic encourages global collaboration and modification, while the use of C++, Python, and Java allows efficient simulations and easy integration with other software tools.

We developed a simulation setup using Geant4 to model the transmission of the pin beam through the phantom material in our experimental setup. The simulation setup considered the optical properties of the phantom material, the geometry of the setup, and the characteristics of the CMOS camera. To simulate the optical processes inside the material and on its surface, we utilized Geant4’s optical physics and optical surface physics modules [23]. We employed Mie scattering to accurately simulate scattering, absorption, and anisotropy within the material, with the flexibility to consider all relevant parameters [14]. The simulation provided reference data that we compared with our experimental results. We implemented comprehensive data acquisition in Geant4 and the acquired data were stored in root-files for further analysis.

### 2.9. Creating the Simulation Setup

Our main aim was to ensure a good fit between the experimental data we gathered and the Geant4-simulation. The simplicity of our setup, without the inclusion of optics or complex geometries, allowed us to create a realistic simulation with minimal unknown factors. Figure 1 provides a visual representation of the setup for better understanding in comparison to the experimental setup.

Figure 4 shows the visualization of a “run” as an example, involving 1000 photons. The visualization of a run is an important part of troubleshooting and effectively demonstrates the radiation behavior of the LED, the spatial distribution of photons, and the strong absorption of the selected aperture system.

### 2.10. Default Parameters

The chosen phantom material came with four relevant optical parameters, the refractive index n, the reduced scattering coefficient ${{µ}^{\prime }}_{s}$, the absorption coefficient ${µ}_{a}$ and the anisotropy factor g. Geant4 is capable of considering all of these parameters, and the scattering and absorption is implemented as a mean free path (MFP). MediLumine provided us with the reduced scattering coefficient ${{µ}^{\prime }}_{s}$ which was transformed to the non-reduced scattering coefficient ${µ}_{s}$, taking the anisotropy factor g into account as follows [24]:(1)$${µ}_{s}=\left(\frac{{{µ}^{\prime }}_{s}}{1-g}\right).$$

The obtained scattering coefficient can now be converted into a MFP for Geant4 using the following formula:(2)$$l_{s}=\frac{1}{{µ}_{s}}cm.$$

And analogous the absorption coefficient:(3)$$l_{a}=\frac{1}{{µ}_{a}}cm.$$

The anisotropy factor g, which quantifies the directional dependence of the scattering, is determined by the relation given below, with θ representing the scattering angle between the incident and scattered directions of light or radiation [24]:(4)$$g=<\cos\theta >=\int_{0}^{\pi }\rho \left(\theta \right)\cos\theta 2\pi \sin\theta d\theta .$$

In Geant4, Mie Scattering is implemented as an approximation, using the Henyey–Greenstein Function [25]:(5)$$\frac{d\sigma }{d\Omega }=\frac{1-g^{2}}{{\left(1+g^{2}-2gcos(\theta )\right)}^{3/2}},$$
where
(6)dΩ=dcos(θ)dϕ.

As mentioned above, g = ⟨cos(θ)⟩ can be seen as the constant determining the angular distribution. Taking this into account the approximation can be expressed as:(7)$$P\left(cos\left(\theta_{0}\right)\right)=\frac{\int_{-1}^{\cos\left(\theta_{0}\right)}\frac{d\sigma }{d\Omega }d\cos\left(\theta \right)}{\int_{-1}^{1}\frac{d\sigma }{d\Omega }d\cos\left(\theta \right)}=\frac{1-g^{2}}{2g}\left(\frac{1}{\left(1+g^{2}-2gcos(\theta_{0})\right)}-\frac{1}{1+g}\right).$$

Therefore,
(8)$$cos\left(\theta \right)=\frac{1}{2g}*\left(1+g^{2}-{\left(\frac{1-g^{2}}{1-g+2g*p}\right)}^{2}\right)=2p\frac{{\left(1+g\right)}^{2}\left(1-g+gp\right)}{{\left(1-g+2gp\right)}^{2}}-1.$$
where p is a uniform random number between 0 and 1.

MediLumine’s characterization considering our desired wavelength of 590 nm and the parameters implemented in Geant4 are described in Table 1:

### 2.11. Data Acquisition

The data were gathered with a so called “Sensitive Detector” in Geant4. The detector was designed considering the resolution, geometry, material and the quantum efficiency gathered from the detailed EMVA datasheet by Basler [18]. The data were stored in root files for further analysis and comparison with the measured experimental data. Geant4 is capable of simulating different optical processes and the implementation of different optical properties. The main process of optical photons in tissue-like materials is scattering [26]. To give us a deeper insight in the occurring physics processes, the simulated platelet itself was designed as a sensitive detector and gathered further data. The acquired type of data in both sensitive detectors was the exact location of the simulated photons, and the number of processes that occurred for each photon. The simulated CMOS-chip was also capable of retrieving the photon time of flight and the traveled distance. For a better understanding of the processes inside the simulated platelet, it was designed to obtain additional information about the angle distribution, the photons scattered forward and backward and the processes that occurred solely on the surface.

### 2.12. Data Comparison

The experimental data obtained from the image analysis were compared with the reference data from the Geant4-simulations. To obtain a deeper understanding of the processes involved in the distribution of photons, and ultimately the formation of a captured image, the different parameters mentioned in Section 2.7 were considered individually and related to the results. Statistical analysis was performed to evaluate the agreement between the experimental and simulated results.

## 3. Results

### 3.1. Images and Acquired Data

The images taken with our experimental setup, shown in Figure 5, were analyzed with python scripts and the radial and line profile were calculated.

The key quantity with which to compare the data is the FWHM values of the aforementioned radial and line profile, and the slope of the radial profile calculated in the range of the FWHM. An example of a comparison is shown in Figure 6, and the results of the different analyzed thicknesses and materials are displayed in the following figures.

The radial profile shows a strong correlation, as expected, indicating the average decrease in attenuation from the center to the edges. In contrast, the line profile, specifically the calculated FWHM values, highlight variations in platelet thickness between the samples. Figure 7, Figure 8 and Figure 9 show the images and plots of different thicknesses and materials.

The presented figures effectively demonstrate the ability to accurately differentiate between individual thicknesses and materials through the utilization of collected parameters. Notably, even the plots in the final figure, which differ solely in the order of placement, exhibit a high level of distinguishability. To ensure maximum reproducibility, the following measures were implemented: a constant current source for the LED that both monitors and records the incoming and outgoing current; a closed aperture system to prevent light entry and exit; and standardized image acquisition with a defined maximum gray value. Depending on the quality of the material, these measures help to maintain the standard error within a single-digit percentage range.

### 3.2. Adapting the Simulation

Upon completion of our experimental setup and data collection, we proceeded to fit our simulation model to the obtained data. The initial step involved ensuring a high level of correspondence between the light source and the shaped pencil beam in both the simulation and experimental settings. To achieve this, simulations were conducted to verify the alignment and make any necessary adjustments. The result of this fitting process is presented in Figure 10.

The plots in Figure 10 illustrate the comparison between the simulation and the experimentally captured image of the isolated pin beam, excluding the platelet. The FWHM values of the line and radial profiles exhibit remarkable agreement with only slight deviations. Both profiles demonstrate reproducible values of 0.63 mm and 1.07 mm, respectively, falling within the anticipated range of the 1 mm pin beam diameter. Additionally, the shape of the plotted profiles exhibits excellent conformity, displaying a sharp decline at the edge of the utilized aperture. These results confirmed the suitability of the experimental setup and allowed for minor adjustments to enhance the simulation’s accuracy in relation to the real-world conditions.

We then proceeded to simulate the behavior of the 2 mm and 3 mm platelets, employing the predefined parameters discussed in Section 2.10. The results of these simulations are presented in Figure 11 and Figure 12.

The comparison between the simulation and experiment shows significant differences in the FWHM, mean error, and overall visual appearance. These differences are in contrast to the expected theoretical agreement, where the simulation is expected to closely approximate reality using the implemented parameters of INO. The simulated signal is significantly more expanded and the integrated drop to the edges is significantly weaker. The consistency of the deviation at different thicknesses of the platelets made an error in the setup or the camera unlikely. Two parameters, the MFP and the anisotropy are responsible for the widening of the signal. A complex fitting process regarding these parameters was therefore necessary and will be discussed in detail below.

### 3.3. Scattering, Absorption and Anisotropy

The objective of this part was to identify the dominant physical process and to determine the optimal parameter for achieving a good fit. Due to the qualitative nature of our measurements, a detailed investigation of absorption was not possible. Simulations were conducted with various ${µ}_{a}$ values across different types of tissue [26] and analyzed, showing no significant impact. Our focus was on the MFP and anisotropy, and we conducted a systematic analysis of the available values, with a specific emphasis on the 2 mm platelet. These findings were then extrapolated to other thicknesses.

#### 3.3.1. Anisotropy Factor g

Changes in the anisotropy factor g towards values close to 1 led to a general decrease in FWHM and a mean error, indicating a good fit. Conversely, a lower factor g does not necessarily result in an increase in FWHM. This is most likely due to the fact that above a certain threshold of processes, the distribution of photons becomes completely isotropic and lower g values no longer have any further influence. This connection is described in more detail below. A comparison of different g factors with the default value of 0.62 is shown in Figure 13 and Figure 14.

To investigate the correlation of FWHM and the factor g, the angular distribution of the scattered photons was collected for different values. Geant4 is able to record the direction vector of each scattering process which was converted to an angle and displayed as a polar plot for better clarity in Figure 15.

Figure 15 clearly demonstrates the impact of the factor g on the angular distribution following the Henyey–Greenstein Phase function where g defines p(cosθ) [27]:(9)$$p\left(cos\theta \right)=\frac{1}{2}\frac{1-g^{2}}{{\left[1+g^{2}-2gcos\theta \right]}^{\frac{3}{2}}}.$$

As the g factor approaches 1, the photons tend to be predominantly directed forward. Consequently, for a constant MFP, fewer processes occur within the platelet. This observation is further supported by Figure 16, as discussed below.

As shown, the g factor has a massive influence on the distribution of the process frequency with a mean number of processes from 38 (factor g = 0.1) to 13 (factor g = 0.9). For comparison purposes, Figure 17 demonstrates the comparable influence on the number of process frequencies when altering only the MFP. In specific scenarios, it becomes challenging to discern whether the observed effect is attributable to an increased anisotropy or a decreased MFP.

#### 3.3.2. Scattering

The next step was a systematic simulation of different lengths of the MFP $l_{s}$, equivalent to the scattering coefficient ${µ}_{s}$. The resulting changes of the FWHM are displayed in Figure 18.

The plot illustrates the strong dependence of the MFP on the size of the FWHM value. Note that the respective experimentally determined FHWM values are partially achieved with either the line profile or the radial profile, but only with a factor g of 0.9 both simultaneously. This is also shown in the plot of the analyzed mean errors in Figure 19.

### 3.4. Final Results

Our systematic investigation allowed us to reliably determine the optimal fitting parameters. It was observed that an MFP of around 0.04 cm or the equivalent scattering coefficient ${µ}_{s}$ of 25 provided the best fit across different values of ${µ}_{s}$ and the anisotropy factor g. Moreover, a factor g of approximately 0.9 exhibited the lowest mean error and resulted in nearly perfect alignment of the calculated FWHM values. The corresponding results are presented in Figure 20, Figure 21, Figure 22 and Figure 23, which show the results for the 2 mm phantom platelet, 3 mm phantom platelet, and the different thicknesses and sequences of the investigated PMMA samples.

The aforementioned findings enabled us to successfully simulate the PMMA material even in the absence of prior knowledge of its optical parameters. By combining PMMA with the phantom material, we were able to further validate our experimental setup and the simulation in the context of more complex processes. This is illustrated in Figure 22 and Figure 23.

As can be seen in Figure 22 and Figure 23, the order in which the materials are arranged in relation to each other alone has a significant influence on the measured signal. Materials with a short MFP additionally amplify the effect of the broadening of materials with a longer MFP.

The obtained simulation results indicate a favorable agreement with the experimental data, although additional investigations are required for a comprehensive understanding. It is evident that the chosen parameters are not coincidentally appropriate, and significant deviations, particularly in the factor g, are observed compared to the given parameters.

Overall, as expected, the same material scatters light more with increasing thickness. The FWHM is broadened, but the general progression of radial and line profiles correlates strongly, given that optical parameters such as scattering length and anisotropy factor remain constant. Once these parameters are determined, they can be effectively extrapolated in simulations and applied in various scenarios (e.g., mixing different materials).

Other materials, like PMMA, exhibit significant differences in both radial and linear profiles, thus allowing for a clear distinction. However, once their parameters are established, they too can be extrapolated to yield accurate simulation results.

Data from the literature show that materials made of TiO_2_ predominantly exhibit an anisotropy factor g in the range of 0.5 to 0.7 [28,29]. However, other studies indicate significant discrepancies between the estimated factor g and the experimentally obtained values [30]. Therefore, we dedicated our efforts to investigating the factors contributing to this divergence and assessing the compatibility of our findings with the default parameters received by INO.

### 3.5. Examining the Given Parameters by MediLumine

The method used for determining the optical parameters of the phantom is related to Time of Flight (TOF) Absorption Spectroscopy, using a 20 mm thick phantom [16,31]. We developed a way to determine the TOF of the photons in our simulation and compared it with various parameters to the results obtained by MediLumine. Figure 24 shows the results distributed by INO.

On the x-axis, one can observe the temporal resolution, while the y-axis represents the photon counts. The photons reach the detector with a time difference (Δt) of 6 ns, and the signal is approximately broadened to 1.5 ns. The collected data can be utilized to determine the average track length of a photon when the velocity of photons (v) is equal to the speed of light (c). As mentioned in Section 3.3.1, the processes and, consequently, the track length vary not only with the MFP but also with the decrease in anisotropy. Our simulation yielded similar results and enhanced our comprehension of the interaction between the MFP and the anisotropy factor. Figure 25, Figure 26 and Figure 27 depict a comparison of different parameters with almost indistinguishable outcomes.

These results show that unambiguous measurement results are very difficult to generate in the interaction of the different optical properties and depend on the thickness of the selected material. Figure 26 shows a difference from deviating MFP of almost sixfold but almost no to little distinguishability in terms of TOF and total number of photons. This led us to the assumption that our simulated parameters and the data distributed by INO are compatible after all if the uncertainties mentioned are considered. In contrast, our experimental setup seems to achieve a distinguishability in terms of anisotropy at a sample thickness of only 2 mm. As already mentioned, the angular distribution of the photons is lost at a certain number of average processes and gives an isotropic image.

## 4. Discussion

Using Geant4 standalone libraries (without GATE extension [14]) for all-optical processes is, to the best of the authors’ knowledge, a novelty and has provided us with valuable insights into the processes that are crucial for the quantification of bioluminescence data. In our simulation framework, we carefully considered key parameters and variables, including the anisotropy factor, scattering coefficient, absorption coefficient, and surface roughness. By accurately modeling the behavior of the light source, in this case, an LED, and simulating the physical processes using Mie scattering, we were able to replicate the emission and distribution of bioluminescent signals in a controlled environment.

Our analysis of signal broadening and distribution revealed several important findings. We observed that signal broadening is heavily dependent on the mean free path, but its interaction with the anisotropy factor g made it challenging to distinguish the individual contributions of these factors. By comparing our simulation results with experimental data, we were able to align the two, indicating the accuracy and validity of our simulation approach.

However, it is important to mention the limitations of our Geant4 simulation approach. One limitation is the particle-based nature of the simulation, which may not fully capture certain optical effects, such as diffraction. Additionally, the complexity of tissue interactions and the qualitative nature of our experimental setup limited our ability to fully address the absorption processes. Future studies could address these limitations by incorporating more comprehensive optical physics models and exploring the interaction of different tissue types.

Researchers and practitioners in these fields can utilize our simulation framework and insights to improve their experimental design and data interpretation. By integrating our simulation results into their studies, they can enhance the accuracy and efficiency of their measurements and gain a better understanding of bioluminescent processes.

Moreover, our study opens avenues for further research and exploration. The interaction between different tissue types and the analysis of bioluminescence using techniques like Time of Flight warrant further investigation. By expanding the scope of the simulation to incorporate factors like the interfaces of different tissue types, we can continue to advance our understanding of bioluminescence and its applications.

In conclusion, our Geant4-based simulation study provides valuable insights into the physical mechanisms underlying signal broadening and distribution in bioluminescence imaging. The simulation framework and gathered findings contribute to the broader scientific community’s understanding of bioluminescence and offer researchers and experimentalists a powerful tool to improve their experimental design and interpretation of data. With continued research and development, bioluminescence can become more quantifiable, leading to advancements in fields such as medical imaging and fundamental biological research.

## Figures and Tables

**Figure 1 sensors-23-06458-f001:**
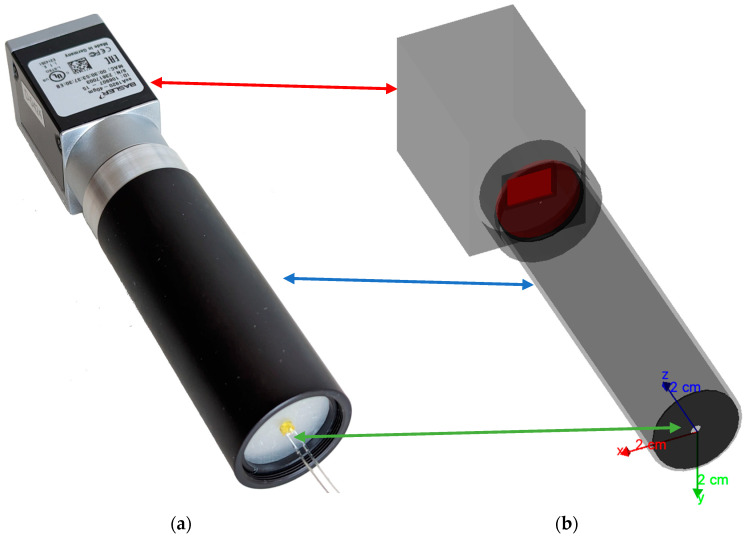
(**a**) Experimental setup: the figure shows the camera (red arrow) on the left and the assembled aperture system attached (blue arrow). The green arrow shows the LED mount with an example LED. (**b**) Analogous visualization of the setup in Geant4.

**Figure 2 sensors-23-06458-f002:**
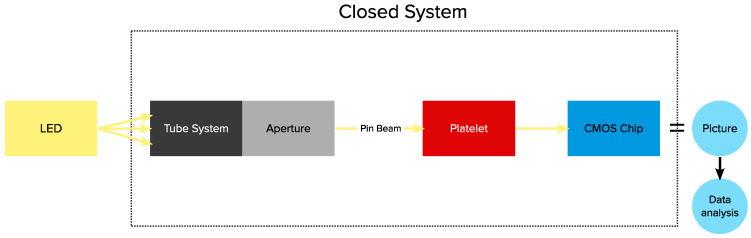
A flowchart to better understand the experimental setup and the path of the light. The LED is placed on the left side, the light is reduced to a 1 mm diameter beam in the tube and aperture system. The signal behind the inserted material is then detected by the CMOS sensor which records images that can later be used for detailed data analysis.

**Figure 3 sensors-23-06458-f003:**
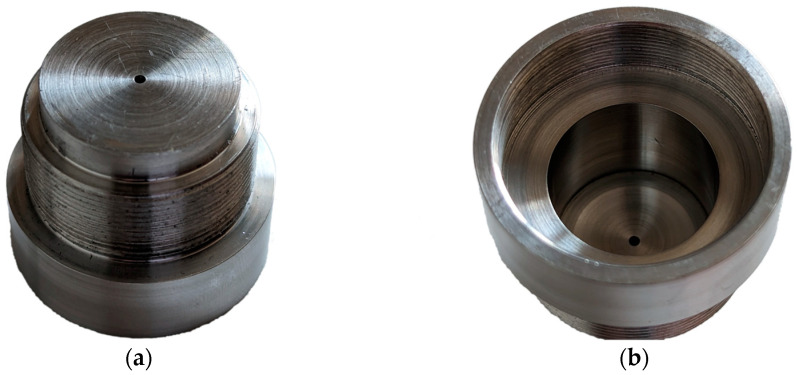
(**a**) Pictures of a custom-made pinhole with integrated thread and spacer ring as well as an aperture of 1 mm, viewed from the front; and (**b**) custom-made pinhole with integrated thread and spacer ring as well as an aperture of 1 mm; viewed from the back.

**Figure 4 sensors-23-06458-f004:**
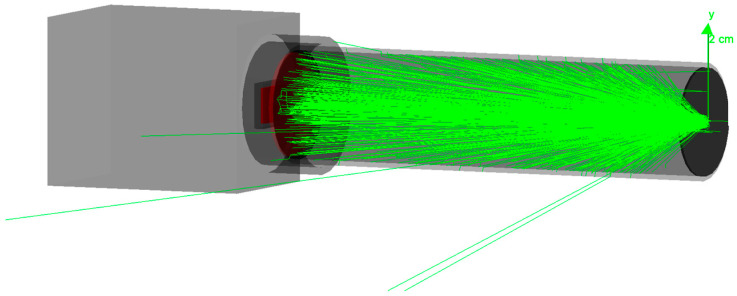
Geant4 visualization of a run with 1000 photons. It shows the photon distribution of the LED on the right side and the absorption of the aperture system, resulting in a pin beam on the left side.

**Figure 5 sensors-23-06458-f005:**
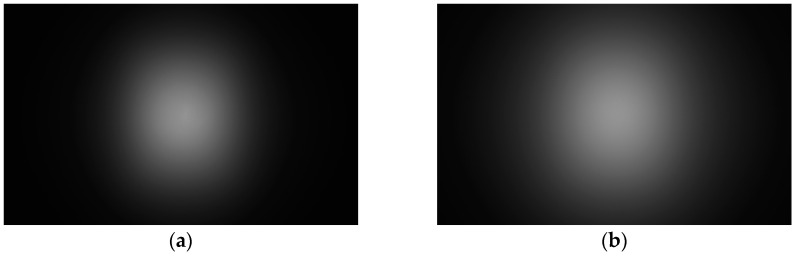
(**a**) This figure shows a monochrome image of the 2 mm phantom material platelet illuminated with the pin-beam, taken with the Basler acA1920-40gm camera; (**b**) this figure shows analogues to (**a**) the monochrome image of the 3 mm phantom material platelet illuminated with the pin-beam, taken with the Basler acA1920-40gm camera. The image shows an overall wider distribution of the initial pin beam, compared with (**a**).

**Figure 6 sensors-23-06458-f006:**
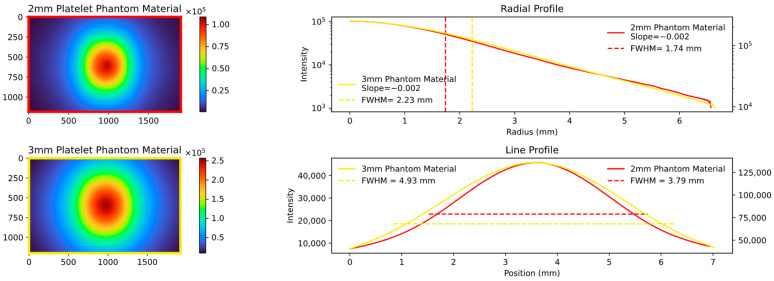
Comparison plot of two experimental setups: the **top left** shows the 2 mm platelet phantom material (analogous to Figure 5a in a converted colormap); the **bottom left** shows the 3 mm platelet phantom material (analogous to Figure 5b). The radial and line profiles are plotted, and the FWHM and slope are calculated. The radial profile exhibits a strong correlation, while the line profile shows differences, as expected when comparing the same material but with different thicknesses. The difference in the line profile and colorbar scale is on the one hand due to the normalization of the two line profiles in order to be able to map them to each other and on the other hand due to the fact that the line profile calculates the average, whereas the colorbar represents the absolute maximum.

**Figure 7 sensors-23-06458-f007:**
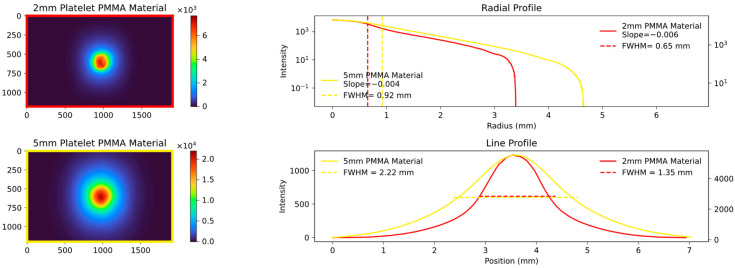
Comparison plot of two experimental setups; the **top left** shows a 2 mm PMMA platelet; the **bottom left** shows a 5 mm PMMA platelet. The radial and line profiles are plotted, and the FWHM and slope are calculated. Analogues to Figure 6, the radial profile exhibits a strong correlation, while the line profile shows differences, as expected when comparing the same material but with different thicknesses.

**Figure 8 sensors-23-06458-f008:**
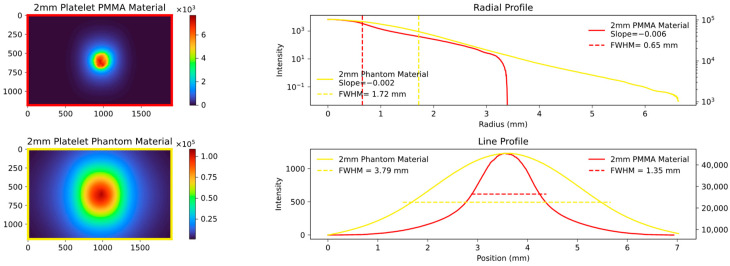
Comparison plot of two experimental setup: the **top left** shows a 2 mm platelet PMMA material, the **bottom left** shows a 2 mm platelet Phantom material. The radial and line profiles are plotted, and the FWHM and slope are calculated. As anticipated, there is no match between the radial and line profiles since we are considering different materials.

**Figure 9 sensors-23-06458-f009:**
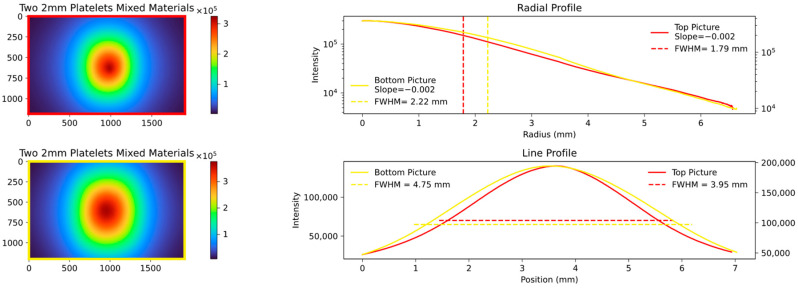
Comparison plot of two experimental setup: the **top** one shows a 2 mm PMMA platelet mixed with a 2 mm phantom material platelet; the **bottom left** shows the same composition but arranged in the opposite direction. The radial and line profiles are plotted, and the FWHM and slope are calculated. The change in material arrangement exhibits a behavior similar to comparing different thicknesses of the same material. The radial profile demonstrates a good visual similarity, while the line profile reveals more noticeable differences. This pattern is also evident in the simulations (Figures 22 and 23).

**Figure 10 sensors-23-06458-f010:**
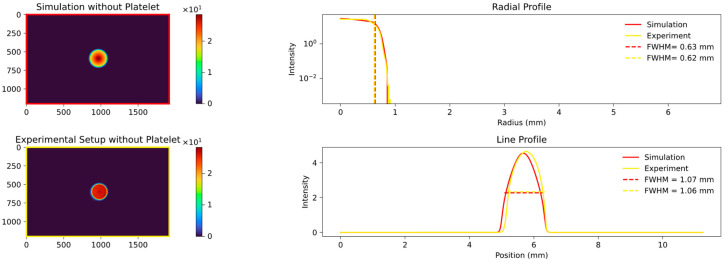
Comparison between experiment and simulation without the platelet. Shown are the simulated light source on the **top left** and the experimental picture at the **bottom left**.

**Figure 11 sensors-23-06458-f011:**
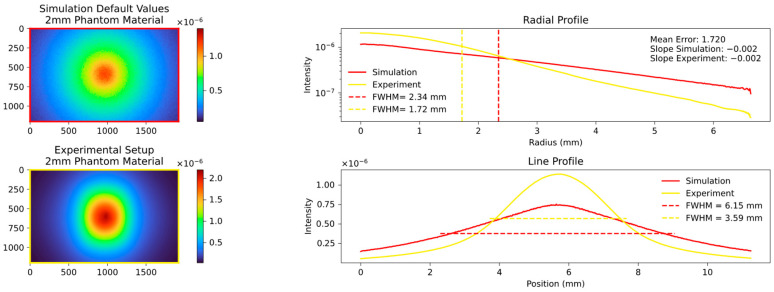
Comparison of simulation and experiment with default parameters for the 2 mm platelet showing poor agreement both visually and in the calculated parameters FWHM, mean error and slope. The simulated signal is almost twice as wide and the slope of the radial profile is less than half of the value of the simulated signal.

**Figure 12 sensors-23-06458-f012:**
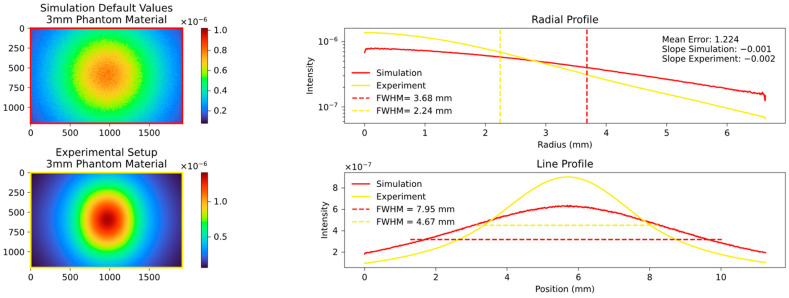
Comparison of simulation and experiment with default parameters for the 3 mm platelet showing a similar poor agreement analogous to Figure 8.

**Figure 13 sensors-23-06458-f013:**
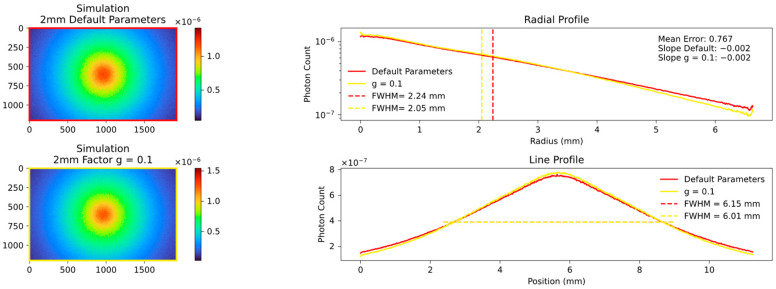
Comparison of two simulations with default factor g = 0.62 on the **top left** and factor g = 0.1 on the **bottom left**. The FWHM, slope and overall visual appearance of both simulations are similar because the average number of optical processes reached a stage where isotropic scattering is to be expected.

**Figure 14 sensors-23-06458-f014:**
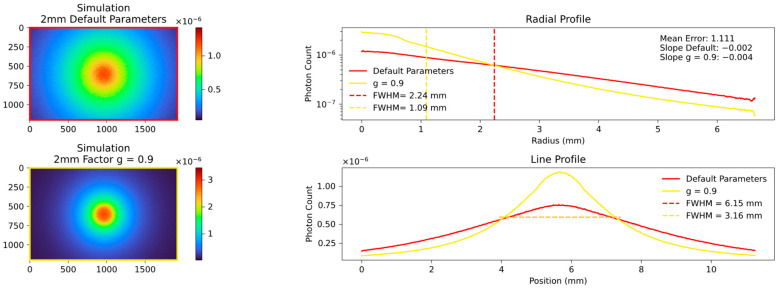
Comparison of two simulations with default factor g = 0.62 on the **top left** and factor g = 0.9 on the **bottom left**. The FWHM, slope and overall visual appearance of both simulations differ greatly because the average number of optical processes is significantly lower at g = 0.9 and the photons are pushed in forward direction. The final fitted results are shown from Figure 21 onwards.

**Figure 15 sensors-23-06458-f015:**
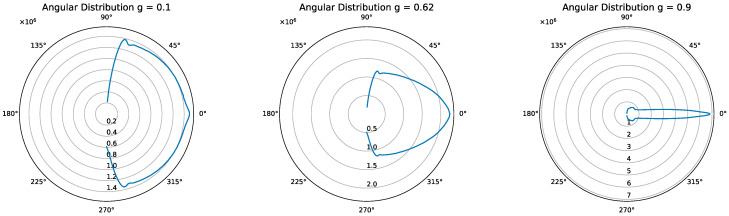
Polar plot of the angular distribution inside the platelet. As displayed, smaller g values result in a wider range in angular distribution, making it completely isotropic after a few scattering processes.

**Figure 16 sensors-23-06458-f016:**
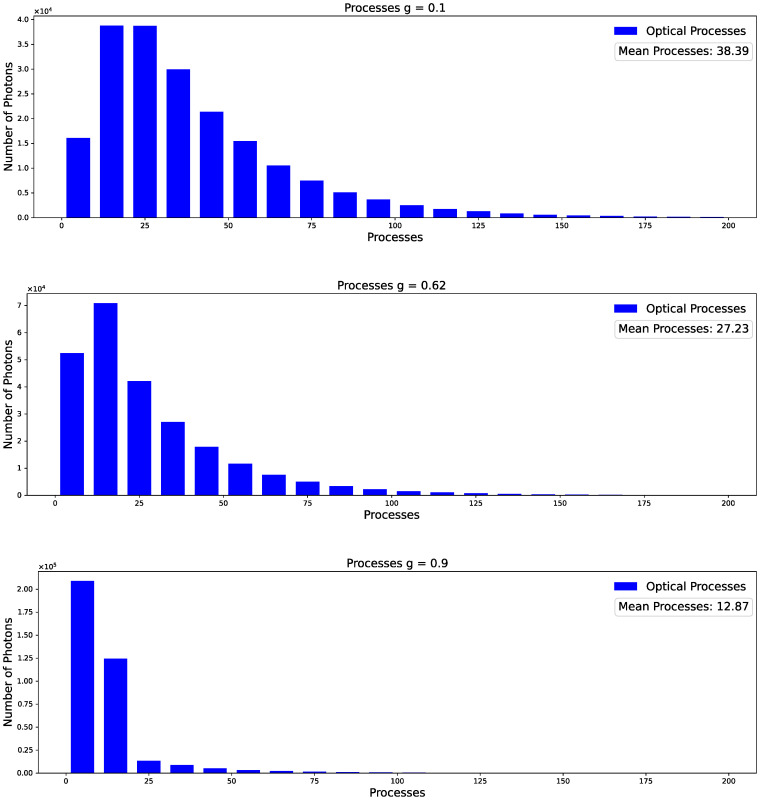
Distribution of the number of processes occurring at different factor g. The plots show a significant decrease in the number of mean processes and a distribution towards lower process frequency for g values approaching 1.

**Figure 17 sensors-23-06458-f017:**
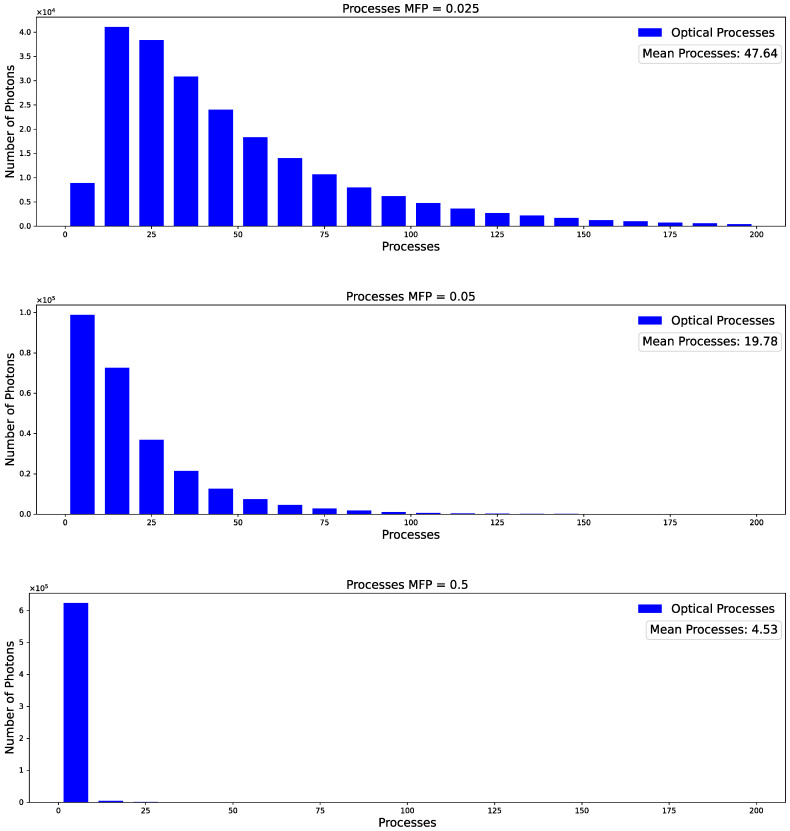
Distribution of the number of processes occurring at different MFP. Analogous to Figure 16, we see a decreased number of mean processes at longer MFP.

**Figure 18 sensors-23-06458-f018:**
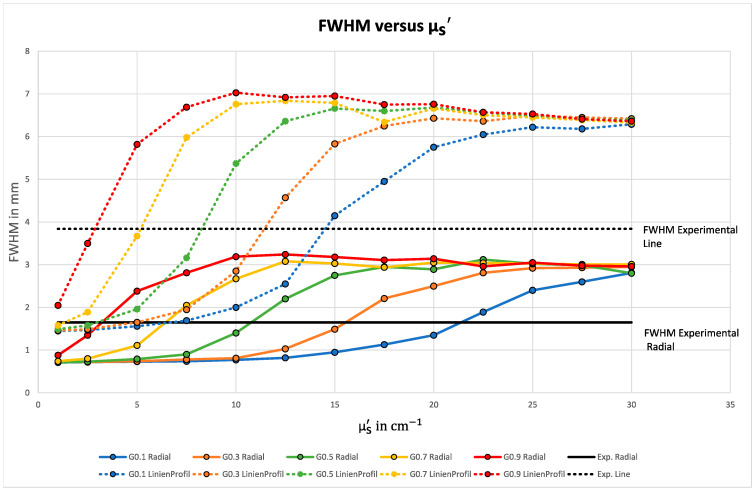
Plot of the FWHM versus ${µ}_{s}^{\prime }.$ The dashed lines show the FWHM of the line profiles and the continuous lines the FWHM of the radial profiles, while the experimentally obtained data are marked in black as a target value. The data marked in red (factor g = 0.9) are the only ones that can simultaneously represent both target FWHM values.

**Figure 19 sensors-23-06458-f019:**
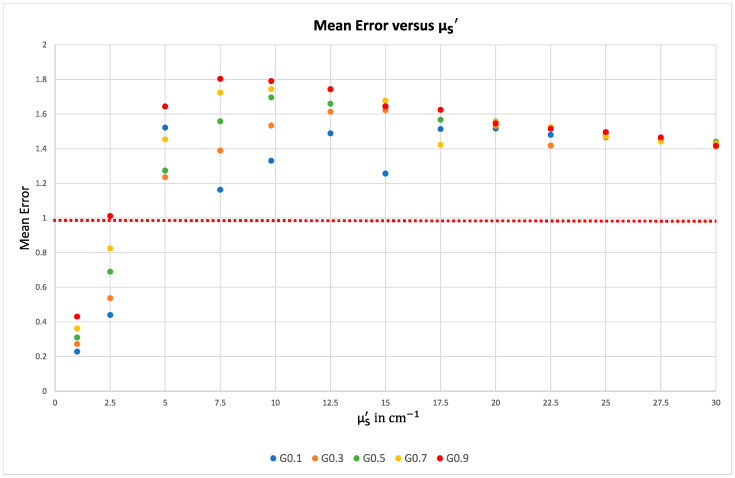
Plot of the mean error versus ${µ}_{s}^{\prime }$. The dashed line represents the target value of 1, indicating perfect agreement between the simulation and the experiment. Only the simulation with a g factor of 0.9 and a ${µ}_{s}^{\prime }$ value of 2.5 approaches this value.

**Figure 20 sensors-23-06458-f020:**
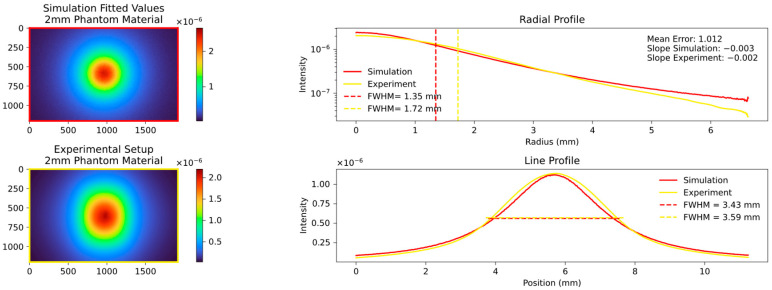
Comparison plot of the experimental setup with the fitted parameter simulation. The **top left** shows the simulation of the 2 mm phantom material platelet with fitted parameters and the **bottom left** shows the experimental setup with 2 mm phantom material platelet. The radial and line profiles are plotted, and the FWHM and slope are calculated. The parameters obtained through fitting, keeping the MFP unchanged but setting the factor g to 0.9, exhibit remarkably strong agreement in all aspects.

**Figure 21 sensors-23-06458-f021:**
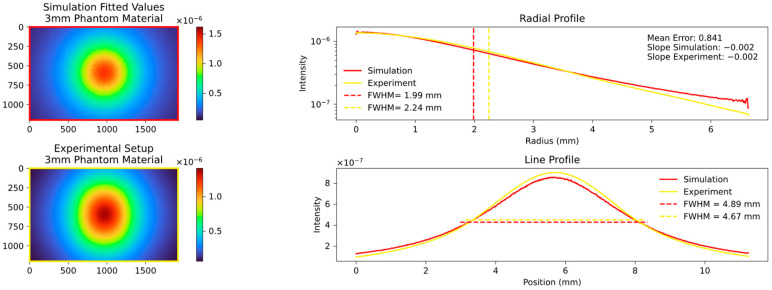
Comparison plot of the experimental setup with the fitted parameter simulation. The **top left** shows the simulation of the 3 mm phantom material platelet with fitted parameters and the **bottom left** shows the experimental setup with 3 mm phantom material platelet. Analogous to Figure 20 the agreement is very good also at a thickness of 3 mm with an unchanged MFP and the factor g = 0.9.

**Figure 22 sensors-23-06458-f022:**
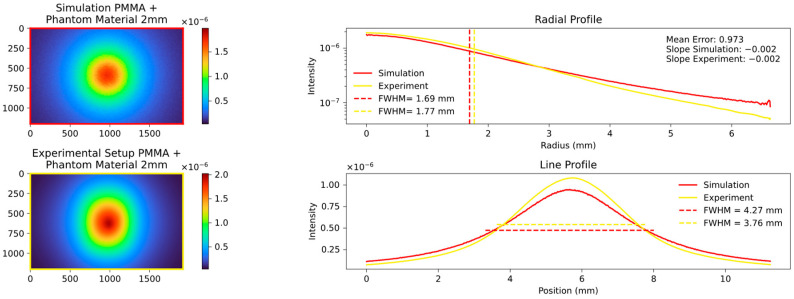
Mixed sequence of phantom and PMMA material, both 2 mm thick.

**Figure 23 sensors-23-06458-f023:**
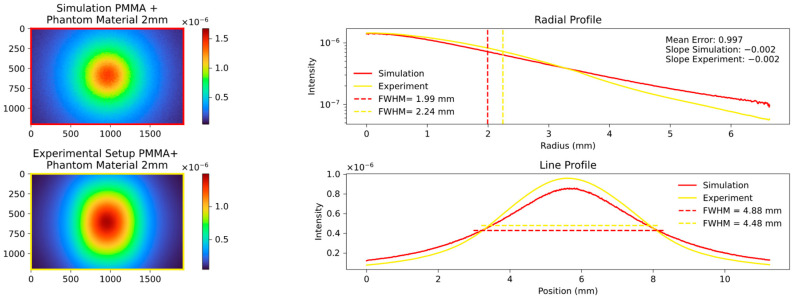
Mixed sequence of phantom and PMMA material both 2 mm, different sequence; analogous to Figure 9, the plots demonstrate differences in the line profile but a strong visual correlation in the radial profile. This observation is also evident in the simulations.

**Figure 24 sensors-23-06458-f024:**
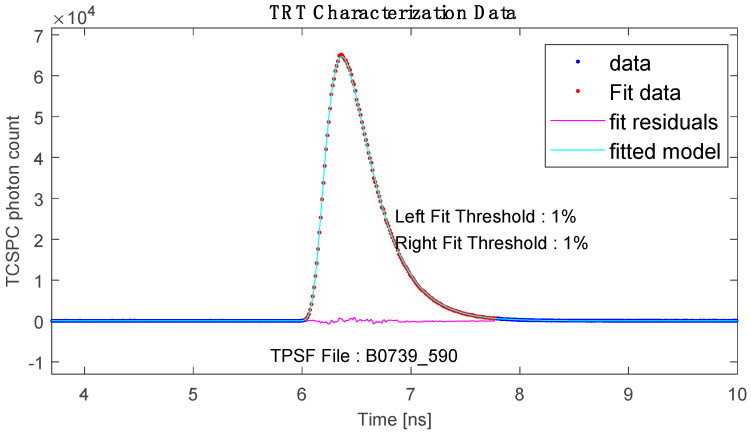
TOF spectroscopy by MediLumine at 590 nm, taken from [32].

**Figure 25 sensors-23-06458-f025:**
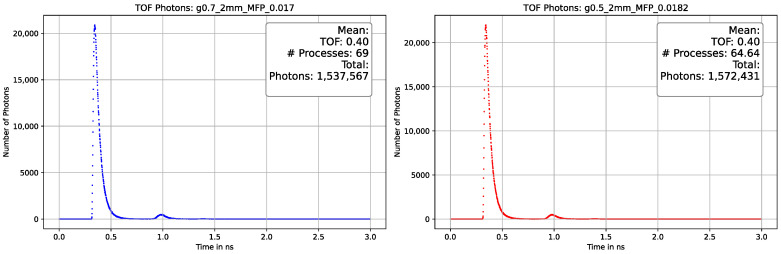
Comparison plot of the photon-TOF with different optical parameters with a 2 mm thickness of the phantom material. The plot on the **left** side is gathered with a factor g = 0.7 and MFP = 0.017, on the **right** side with a factor g = 0.5 and MFP = 0.0182. The TOF is identical in the sub-nanosecond range and the deviation in the total number of photons is in the lower single-digit percentage range. A second peak can be seen in the range of 1 ns, caused by photons scattered first back and then forward again, which then cover twice the path distance.

**Figure 26 sensors-23-06458-f026:**
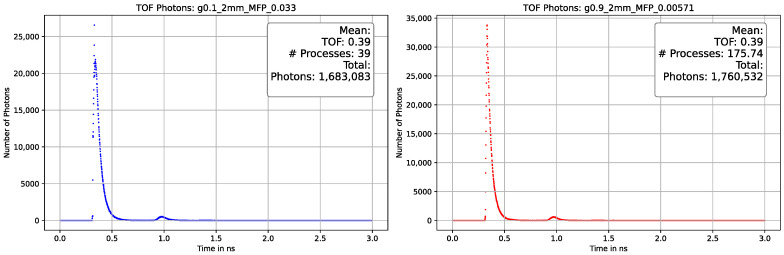
Comparison plot of the photon-TOF with different optical parameters with a 2 mm thickness of the phantom material. The plot on the **left** side is gathered with optical parameters factor g = 0.1 and MFP = 0.033, and on the **right** side with factor g = 0.9 and MFP = 0.00571. The TOF is identical in the sub-nanosecond range and the deviation in the total number of photons is in the lower single-digit percentage range. A second peak can be seen in the range of 1 ns, caused by photons scattered first back and then forward again, which then cover twice the path distance.

**Figure 27 sensors-23-06458-f027:**
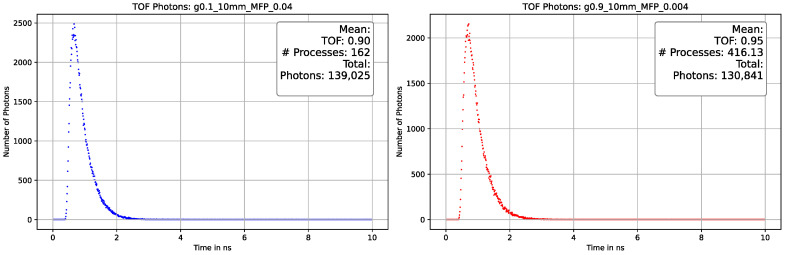
Comparison plot of the photon-TOF with different optical parameters with 10 mm thickness. The plot on the **left** side is gathered with optical parameters factor g = 0.1 and MFP = 0.04, and on the **right** side with factor g = 0.9 and MFP = 0.004. The TOF deviates in the sub-nanosecond range and the total number of photons in single-digit percentage range. The second peak disappeared due to the greater thickness of the material and the associated significantly more frequent scattering processes that completely diminished the signal.

**Table 1 sensors-23-06458-t001:** Parameters used and the equivalent implementation in Geant4.

Parameter	Default Value at 590 nm	Geant4 Equivalent	Geant4 Parameter
$l_{s}$ = ${µ}_{s}$	25.82 ${cm}^{-1}$	MIEHG	0.03873 cm ^2^
$l_{a}$ = ${µ}_{a}$	0.131 ${cm}^{-1}$	ABSLENGTH	7.63 cm ^2^
g	0.62	MIEHG_FORWARD	0.62
n	1.511	RINDEX	1.511
Ratio	nan	MIEHG_ FORWARD_RATIO	1
${{µ}^{\prime }}_{s}$	9.86 ${cm}^{-1}$	nan ^1^	nan ^1^

^1^ No option in Geant4 for the reduced scattering coefficient. ^2^ Implemented in Geant4 as MFP.

## Data Availability

The data presented in this study are available on request from the corresponding author.

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
