# Peer review of "Quantitative Assessment on Optical Properties as a Basis for Bioluminescence Imaging: An Experimental and Numerical Approach to the Transport of Optical Photons in Phantom Materials"

_sensors, 2023, doi:10.3390/s23146458_

Round 1

Reviewer 1 Report

Review report

Manuscript ID: sensors-2457197

This manuscript reported the result of the experiments comparing the measurement and computer-simulated light intensities for a future application to bioluminescence imaging. The measurement was done with a simple CMOS camera with an aperture. The numerical simulation of the light intensities employed Geant4 which is an open source provided via CERN website.

The experimental conditions in the manuscript seem not close to those in the practical bioluminescence imaging. Use of Geant4 to simulate the light propagation in biomedical optics is not so popular and maybe unique. If the authors found some advantages of Geant4 for the biomedical optics, it might be worth to be published. However, the manuscript does not describe the differences between Geant4 (based on Mote Carlo method basically) and the other Monte Carlo-based method popular in biomedical optics. The reviewer feels that the authors just tried Geant4 provided. This manuscript lacks novelty unfortunately. From the manuscript, the reviewer feels that the authors work is now being prepared for a next step in experimental and computational viewpoints.

The reviewer agrees that the quantification for bioluminescence imaging is very important. To achieve the quantification, it must be essential how to reconstruct the tomographic image of bioluminescence sources inside mouse. Many efforts to reconstruct the tomographic image (BLT, FMT) to quantify the optical properties and light sources inside living body noninvasively has been reported in biomedical diffuse optics field. The reviewer thinks that the efforts, which must not be ignored in Introduction, can suggest the future directions of the authors’ studies. The reviewer expects further progress in the authors woks with respect.

Reviewer 2 Report

The manuscript by Tim Karstens et.al. is devoted the study of bioluminescence imaging towards more quantitative measurements. The manuscript is well written and I do think it will be interesting for Sensors readers. Authors made a good amount of experiment and calculations for data validation. In simulation framework, authors considered several key parameters and variables and made the model of the behavior of the LED light source and simulated the physical processes.

I have minor comments only:

1. The Figures numbering went wrong, please, correct.

2. Figures 21-23 - axis titles are written in German

3. Authors should provide errors (accuracy) of the experimental results (for example for FWHM on Figures 7,8).

4. Can authors please explain why the position of the maximum of the simulation differs from the position of the maximum of the experiment (Fig. 7, line profile). The same picture: line profile intensity axis changes from 0 to ~1300, however, on the left side intensity changes 0 to ~6000 for 2mm platelet PMMA material and from 0 to ~20000 for 5mm platelet PMMA material. What is the reason for such a change of intensity in the data presentation? The same question for Fig.8.

5. Can authors please write a few sentences in the manuscript about potential changes in the result of mixed sequence of phantom - does the order matter? I do believe that after reading the article, readers should not be left with any doublethink about this. 

Reviewer 3 Report

This manuscript described an experimental setup and Geant4-simulations to gain a better understanding of the optical properties and processes involved in bioluminescence imaging. This is a very systematic work, but there are some details that need further explanation.

1.    In the introduction section, the authors should introduce the recent progress of bioluminescence imaging. Explain what issues will be addressed in this job? What are the innovative points compared to existing technology?

2.    Figure1 to Figure 3 can be used as a comprehensive schematic diagram. Please carefully check for some clerical errors. For example, the label of the second image is Fig.2, not Fig.5.

3.    In the section of 2.5, please provide detailed basic parameters such as the center wavelength and half width of the LED light source.

4.    In the results section, there should be some textual description of the main conclusions for the figures. For example, what is the final trend of change caused by different thicknesses and materials?

Reviewer 4 Report

The present work deals with experimental and numerical simulation of bioluminescence imaging. However, the manuscript as in its present form needs improvments. In the following are some comments:

1) More technical data have to be added about all used instrumentation as well the used software.

2) A flowchart for the experimental setup can added to be more clear.

3) For the simulation part, some additional data have to be given. What are the equations describing the physical phnomena? How these equations are implemented? What is the used numerical method? ...

4) The equations must be numbered

English should be revised.

Round 2

Reviewer 1 Report

The reviewer thanks the authours for the authors' reply and modifitied mascript. However, the reviewer still can not understand why the manuscript can be published.

The reviewer undersatands that the manuscript describes that the authors failed to simulate the detected light intensity for the phantom of which the optical propeties were given by manufacturer. Then the authors made up the optical properties of the phantom, which were not agree with the given ones, to fit their Geant4 simulation to measurement.

There can not be found any sufficient discussions about the error in Geant4 simulation and/or the validity of the estimated optical properties by use of Geant4. The reviewer thinks that the story described in the manuscript does not scientifically sound.

It is better for the authors to reexamin the experiments and simulations again more carefully and to find better way to demonstrate the validity and effectiveness of the authors' method.

In diffuse optical biomedical image field, the problem on scattering for light propagation the authors dealt with are well studied. To demonstrate the advantage of Geant4, another approach should be taken. 

After that, the manuscript, in which the purpose is unclear and methods and results were not well structured, should be reorganized to clarify the important point of the manuscript.

The title should focus on the contents of the manuscript. The current title does not appropriately describe the contents. The authors have to check the sentences and words more carefully. Professional editing can help.   

Reviewer 3 Report

The author basically answered my questions and agreed to accept it.

Author Response

Thank you for your positive review, please note that we changed the Title and added some details to the Introduction to express the abstract nature of our experiment and the investigations.